# Sex Differences in the Efficacy of Immune Checkpoint Inhibitors in Neoadjuvant Therapy of Non-Small Cell Lung Cancer: A Meta-Analysis

**DOI:** 10.3390/cancers15184433

**Published:** 2023-09-06

**Authors:** Guillermo Suay, Juan-Carlos Garcia-Cañaveras, Francisco Aparisi, Agustin Lahoz, Oscar Juan-Vidal

**Affiliations:** 1Biomarkers and Precision Medicine Unit, Health Research Institute La Fe, Av. Fernando Abril Martorell, 106, 46026 Valencia, Spain; guillermo_suay@iislafe.es (G.S.); juancarlos_garcia@iislafe.es (J.-C.G.-C.); francisco_aparisi@iislafe.es (F.A.); 2Medical Oncology Department, La Fe University and Polytechnic Hospital, Av. Fernando Abril Martorell, 106, 46026 Valencia, Spain

**Keywords:** immune checkpoint inhibitors, NSCLC, neoadjuvance, sex, PD-L1, immune system

## Abstract

**Simple Summary:**

Immune checkpoint inhibitors (ICIs) have transformed the treatment paradigm for metastatic non-small cell lung cancer (NSCLC) patients (IB-IIIA) with no targetable driver mutations. Although genetic and physiological factors could suggest a priori differences in response to ICIs regarding sex, there are few works addressing it, and the available results are confusing. It is well established that women have a more proficient immune system; thus, a higher immune editing level is needed to develop metastatic disease, which could explain their better responses in the early phases of disease. Furthermore, the encouraging results observed for metastatic disease have promoted the use of ICIs as neoadjuvant treatments. Here, we aimed to first review the landscape of current neoadjuvant settings used in resectable NSCLC patients, before analyzing whether sex may be a factor that modulates responses to ICIs. To this end, we have carried out a meta-analysis of the available data.

**Abstract:**

Non-small cell lung cancer (NSCLC) is one of the world’s leading causes of morbidity and mortality. ICIs alone or combined with chemotherapy have become the standard first-line treatment of metastatic NSCLC. The impressive results obtained have stimulated our interest in applying these therapies in early disease stage treatments, as neoadjuvant immunotherapy has shown promising results. Among many of the factors that may influence responses, the role played by sex is attracting increased interest and needs to be addressed. Here, we aim to first review the state of the art regarding neoadjuvant ICIs, whether they are administered in monotherapy or in combination with chemotherapy at stages IB-IIIA, particularly at stage IIIA, before analyzing whether sex may influence responses. To this end, a meta-analysis of publicly available data comparing male and female major pathological responses (MPR) and pathological complete responses (pCR) was performed. In our meta-analysis, MPR was found to be significantly higher in females than in males, with an odds ratio (OR) of 1.82 (95% CI 1.13–2.93; *p* = 0.01), while pCR showed a trend to be more favorable in females than in males, but the OR of 1.62 was not statistically significant (95% CI 0.97–2.75; *p* = 0.08). Overall, our results showed that sex should be systematically considered in future clinical trials settings in order to establish the optimal treatment sequence.

## 1. Introduction

Non-small cell lung cancer (NSCLC) is one of the main causes of cancer death worldwide, which is attributed to its late stage diagnosis [1,2,3]. Immune checkpoint inhibitors (ICIs) have emerged as a new treatment option in those patients without target mutations. Anti-programmed death receptor-1 (PD-1), anti-programmed death ligand-1 (PD-L1) and anti-cytotoxic T-lymphocyte protein 4 (CTLA4) have been developed to inhibit immune checkpoint pathways in order to prime anti-tumor activity of cytotoxic T-cells. PD-1 is a checkpoint protein and a composition of the CD28 family [4]. It pertains to a family of suppressor T-cell receptors, which is also expressed in B cells, monocytes and dendritic cells [5]. PD-L1 is a type 1 transmembrane glycoprotein of the B7 ligand family, which is not only expressed on activated B cells and T cells, but also on other type of cells [6]. This pathway intervenes to downregulate T cell functions in antigen-presenting cells. T cells recognize tumor cells and kill them, but when tumor cells upregulate the PD-L1 protein, it binds to PD-1 and leads to the apoptosis of T cells [7]. PD-1 and PD-L1 inhibitors interdict the combination between PD-1 and PD-L1 and effectively produce an activation of depleted immune cells, triggering an immune response to the tumor [8]. CTLA4 is a critical immune checkpoint expressed on the surface of activated T cells. It plays a role in early T cell response development. There is a competition for B7 ligands expressed on antigen-presenting cells against CD28. CTLA4 blocks the formation of the immunological synapse between the T cell and the antigen-presenting cell [9]. The role played by this checkpoint involves maintaining self-tolerance and preventing autoimmune reactions. CTLA4 plays a pivotal role as a regulator of the cancer immunity cycle, and the inhibition of this element has shown to lead to an improvement in the immune response to different tumors [10]. The safety profile of ICIs is acceptable in monotherapy or in combination with chemotherapy. The toxicity is different to that of chemotherapy and associated with hyperstimulation of the immune system, including the reactivation of previous autoimmune syndromes [11,12]. All types of autoimmune diseases have been described, with a preponderance of dermatologic, gastrointestinal, or endocrinological alterations, although clinically significant toxicities are infrequent, and they usually occur in less than 10% of patients [13].

The use of anti-PD1/PD-L1 monotherapy in patients with high tumor PD-L1 expression (≥50%) increased progression-free survival (PFS) and overall survival (OS), as shown in EMPOWER-Lung 1 [14], KEYNOTE-024 [15] and Impower110 [16] trials, which established single-agent immunotherapy as the standard first-line therapy for metastatic NSCLC patients without targeted alterations and high tumor PDL1 expression (≥50%). In contrast, when negative or low PD-L1 expression is observed, the combination of chemotherapy plus anti-PD1/PD-L1 with or without anti-CTLA4 is used, which delivers an increase in both PFS and OS, as shown in the KEYNOTE-189 [17], KEYNOTE-407 [18], IMpower150 [19] and CheckMate 9LA [20] trials. Despite the important clinical benefits reported, the results of phase III clinical trials using ICIs in monotherapy suggested that sex may impact treatment outcomes after observing a worse hazard ratio (HR) for females compared to males [14,15,16]. These results need to be further confirmed, as sex is neither usually considered to be a stratification factor in clinical trials nor included in the clinical guidelines [21,22].

The proven efficacy of ICIs in metastatic disease [17,18,19,20] has promoted their neoadjuvant use in monotherapy or in combination with chemotherapy in different clinical settings. The growing number of settings using ICIs before surgery motivated us to first review the current landscape of the neoadjuvant use of ICIs and evaluate whether sex may have an impact on responses to ICIs in localized disease as it has been shown in the metastatic setting. The main objective of this review and meta-analysis is to review the current landscape of the neoadjuvant treatment of NSCLC after the appearance of ICIs and evaluate sex as a response factor to ICIs in this setting by comparing the data from phase II/III clinical trials in which sex data are available. Besides gaining these results, our secondary objectives are as follows: (1) to suggest treatment monotherapies/combinations in future neoadjuvant trials and (2) recognize the importance of sex as a main factor when selecting a treatment for NSCLC without targetable mutations.

## 2. Methods

### 2.1. Literature Review

To complete this review, we conducted a literature search in the following databases to identify relevant articles: PubMed, Clinicaltrials.gov and Google Scholar. In addition, the latest evidence was considered by including data recent conferences, such as the 2023 American Society of Clinical Oncology, the 2022 European Society for Medical Oncology and the 2022 International Association for the Study of Lung Cancer. The key terms used in the search equations were as follows: “non-small-cell lung cancer”, “NSCLC”, “immune checkpoint inhibitors”, “ICI”, “immunotherapy”, “neoadjuvant treatment”, “preoperative immunotherapy”, “immune sex-based differences”, “cancer sex differences”, “gender differences NSCLC treatment” and “sex-based immunotherapy response”. After obtaining all relevant results, these results were filtered to only retain data derived from clinical trials associated with the purpose of this analysis. The studies were assessed via the preferred reporting items for systematic reviews and meta-analysis (PRISMA) method.

### 2.2. Cohort

NSCLC patients without targetable mutations and stage II/III according to the 8th Edition of TNM in Lung Cancer of the International Association for the Study of Lung Cancer classification who were enrolled in phase II/III trials in which ICIs were used were considered in this review [23]. Trials that used ICIs in monotherapy or combined with other ICIs (e.g., combination of anti-PD(L)1 with anti-CTLA4) and trials that employed ICIs with chemotherapy were selected. Major pathological response (MPR) and pathological complete response (pCR) were reported where available [24,25,26,27,28,29,30,31,32,33,34,35]. Sex was selected where available in each reported trial in order to perform an exploratory meta-analysis to analyse sex as a factor. We found that one bias of this study was that sex was not taken into account as a main stratification factor, meaning that population homogeneity differed between studies.

### 2.3. Statistical Analysis

MPR and pCR stratified based on sex were used as the main endpoints. Since 3 out of 4 studies and 2 out of 4 studies, respectively, were single-arm clinical trials, we performed meta-analyses using non-comparative binary data related to the MPR and pCR. We used the Cochrane Review Manager Software v.5.4.1 (Cochrane, London, United Kingdom) to conduct the statistical analysis. The X2 test and I2 tests were used to evaluate the heterogeneity of the studies. If the heterogeneity was significant, the random effects model was used; if it was non-significant, the fixed effects model was used. Odds ratio (OR) and 95% CI were the effect measures. Significance was fixed at *p* < 0.05.

## 3. Current Clinical Settings of Neoadjuvance in NSCLC

### 3.1. The Neoadjuvant Setting in the Pre-ICIs Era

The good results achieved in the metastatic setting have promoted the use of ICIs as perioperative treatments. The 5-year OSs of patients who underwent a pulmonary resection are 68%, 60%, 53% and 36% for stages IB, IIA, IIB and IIIA, respectively [36]. The new ICIs approaches exploring the neoadjuvant and adjuvant settings aimed to improve the quality of life and OS of these patients. The current use of neoadjuvant chemotherapy in NSCLC is mainly supported by the Meta-Analysis Collaborate Group, which combined 15 randomized controlled clinical trials, showing a significant benefit of pre-operative chemotherapy on survival (HR 0.87; CI 95% 0.78–0.96, *p* = 0.007) and a 13% decrease in the relative risk of death. These findings represented an absolute survival improvement of 5% after 5 years [37]. After comparing these results to those of the adjuvant trials, this setting has been relegated to specific cases included in the main guidelines [22]. Despite these encouraging results, the influence of sex on the neoadjuvant setting has not been deeply considered. Only one review that included a retrospective data analysis of NSCLC (I–III) patients who received neoadjuvant chemotherapy between 2004 and 2014 showed that female sex was a favorable factor in terms of improving responses to treatment [38].

### 3.2. The Development of the Neoadjuvant Setting in the ICIs Era

#### 3.2.1. Neoadjuvant ICI

The use of neoadjuvant ICIs is mostly supported based on the assumption that this treatment may control micrometastases in early phases. In this sense, it has been shown that T cells are activated via the recognition of the presented tumor antigens and travel through the lymphatic system and the bloodstream to reach primary and metastatic sites. Therefore, it has been assumed that immunotherapy may better control the tumor in the pre-operatory setting because of lymphatic and blood flow integrity between the tumor and regional lymph nodes, which is not present in the adjuvant setting [39]. Pre-clinical tests have shown that mice with neoadjuvant ICI therapy had longer survival than those that were treated with adjuvant ICIs [40].

Based on these results, ICIs have been tested in the neoadjuvant setting in monotherapy (Table 1) or combined with chemotherapy (Table 2). In 2018, Forde et al. published a pilot trial of nivolumab in resectable NSCLC [24]. In this study, patients with stage I-IIIA NSCLC received two doses of nivolumab every 2 weeks, followed by surgery in week 4 after the first dose. In total, 22 patients were enrolled, of whom 21 were eligible for the study. The primary endpoints of the study were safety and feasibility; pathological response, expression of PD-L1, mutational burden and neoantigen-specific T-cell responses were also analyzed. The pathological assessment of the removed tumors showed a major pathological response (MPR; defined as ≤10% viable malignant cells) in nine patients (42.85%). The side effect profile was acceptable and not associated with delays in surgery. The NEOSTAR study was a randomized phase II trial of neoadjuvant nivolumab vs. nivolumab and ipilimumab in operable NSCLC (stage I to IIIA) [25]. In this trial, the primary endpoint was MPR, which was met in the nivolumab plus ipilimumab arm (MPR 38%; 8/21 patients). However, it was not reached in the nivolumab arm (MPR 22%; 5/23 patients). The difference between MPR in both arms was not statistically significant. Grade 3 or worse immune side effects were similar between both arms, being 13% in the monotherapy arm and 10% in the combination arm. The LCMC3 study was the largest neoadjuvant ICI monotherapy study. It was a single-arm phase II study in which 181 patients with untreated stage IB-IIIB NSCLC received two doses of neoadjuvant atezolizumab monotherapy [26]. The primary endpoint was MPR, which was achieved in 20% of patients. As a secondary objective, the 3-year survival rate was 80%. In total, 20 patients (11%) suffered G3 or worse immune-related side effects.

Two studies have shown conflicting results: one clinical trial was finished early due to 90-day post-operative mortality [27], and another trial was negative [28]. The first trial, which was named the IONESCO study, consisted of a phase II trial in which durvalumab was administered to patients with resectable stage IB >4 cm-IIIA NSCLC [27]. Three doses were administered on days 1, 15 and 29, followed by surgery at 2–14 days after the last infusion. The primary endpoint was the rate of complete surgical resection; secondary endpoints were safety, OS, PFS, time between the first infusion and surgery, response rate (RECIST 1.1) and MPR. Among the 46 operated patients, 4 deaths occurred due to post-operative complications, although they were not related to durvalumab therapy (3 out of 4 deceased patients had cardiovascular comorbidities). There was not any MPR in this study. The second trial named the PRINCEPS was a phase II study in which a single dose of atezolizumab was administered to resectable stage IA >2 cm-IIIA NSCLC [28]. Surgery was performed at 21–28 days after administration. The primary endpoint was response via RECIST 1.1 and MPR. In total, 30 patients were enrolled, of whom 29 had a R0 surgery, but none of them had MPR. The investigators argued that the negative results could be attributed to the short delay between atezolizumab administration and the surgery.

#### 3.2.2. Neoadjuvant ICI Plus Chemotherapy

After observing the good results of the combination of ICI plus chemotherapy in stage IV NSCLC treatment, recent clinical trials have investigated this approach in the neoadjuvant setting, providing promising results (Table 2) [29,30,31,32,33,34,35]. One of the first studies, which was published by Shu et al., was a phase II trial in which patients with resectable stage IB-IIIA NSCLC received pre-operative treatment with atezolizumab on day 1 and nab-paclitaxel on days 1, 8 and 15 of 21 [29]. Patients received four cycles before proceeding to surgery. The primary endpoint was MPR. In total, 29 of the 30 patients were enrolled and, thus, received an operation. MPR were observed in 17 patients (57%). Side effects were manageable, with no treatment-related deaths occurring. The SAKK 16/14 study was a phase II trial that explored the neoadjuvant setting in stage IIIA-N2-positive patients with NSCLC [30]. It was a single-arm trial in which patients received cisplatin plus docetaxel on day 1 every 3 weeks for three cycles, followed by two doses of durvalumab every 2 weeks. Patients then proceeded to surgery, and one additional year of durvalumab was administered. The primary endpoint was event-free survival (EFS), and the key secondary endpoints were OS, objective response rate (ORR) after neoadjuvant chemotherapy and immunotherapy, pCR and MPR. In total, 67 patients were analyzed, and 55 patients were resected; the 1-year PFS was 73% (95% CI 63–82%), which met the proposed hypothesis. Moreover, 34 (62%) patients achieved MPR, and 10 (18%) patients achieved CPR. In total, 59 (88%) patients had side effects G ≥ 3, and 2 patients died, though investigators did not relate these deaths to treatment. Side effects of special interest regarding durvalumab were hepatic function abnormalities, which were recorded in 6 (10%) patients, and pneumonitis, which was recorded in 2 (3%) patients. The phase II NADIM study explored neoadjuvant treatment in resectable stage IIIA NSCLC patients [31]. Forty-six patients received treatment with carboplatin, paclitaxel and nivolumab on day 1 of every 21-day cycle for three cycles, followed by surgery and adjuvant nivolumab for 1 year. The primary endpoint was PFS at 24 months, and secondary endpoints were OS at 3 years, pathological and imaging response assessed per RECIST v1.1, the proportion of patients who achieved tumor downstaging, the proportion of patients who achieved tumor downstaging, the proportion of patients who had complete resection, the surgical outcome, toxicity and the toxicity profile of the combination. At 24 months, PFS was 77.1% (95% CI 59.9–87.7%), and 34 out of 41 resected patients (83%; 95% CI 68–93%) had MPR. Moreover, 30% of patients had adverse effects of G3 or worse, but none of them were associated with surgery delays or deaths.

Three randomized neoadjuvant trials have been published: the Checkmate-816 [32], the NADIM-II [33,34] and the KEYNOTE-671 [35] trials. The Checkmate-816 study [32] was the only phase III trial published, as well as the largest trial. Patients with resectable IB-IIIA NSCLC were randomized 1:1 to receive either nivolumab plus platinum-based chemotherapy or platinum-based chemotherapy alone, followed by surgery. The primary endpoints were EFS and pCR. The key secondary endpoint was OS. In total, 179 patients were enrolled in each arm of the study. The median EFS was 31.6 months (95% CI 30.2-NR) with the chemotherapy–nivolumab combination versus 20.8 months (95% CI 14–26.7 months) with chemotherapy alone (HR 0.63; 95% CI 0.43–0.91). Median OS was not reached in either the chemotherapy–nivolumab group or the chemotherapy alone groups (HR for death 0.57; 99.67% CI 0.30–1.07). The percentage of patients with MPR was 36.9% in the chemotherapy–nivolumab arm versus 8.9% in the chemotherapy arm, and the pCR was 24% (95% CI 18–31%) in the experimental arm vs. 2.2% (95% CI 0.6–5.6%) in the control arm (OR 13.94; 99% CI 3.49–55.75). Grade 3 or greater treatment-related adverse events occurred in 33.5% of the patients in the experimental group versus 36.9% in the control group. The incidence and characteristics of immune adverse events in the pembrolizumab group were similar to those stated in previous reports. These results have led to FDA approval of this combination.

The NADIM-II study [33,34] was a randomized phase II trial in which patients with stage IIIA NSCLC were randomized 2:1 to receive a combination of carboplatin, paclitaxel and nivolumab on day 1 of every 21-day cycles for 3 cycles versus carboplatin and paclitaxel on day 1 of every 21-day cycles for 3 cycles. Patients then were operated on and received adjuvant nivolumab for 6 months in the experimental arm versus observation in the control arm. The primary endpoint was pCR in the intention to treat population. The secondary endpoints were MPR, the portion of delayed/canceled surgeries, the length of hospital stays, the surgical approach, the incidence of AE related to surgery, safety and tolerability, OS, PFS and potential predictive biomarkers. In total, 57 patients were valid in the experimental arm, of whom 53 patients (93%) proceeded to surgery. Moreover, 29 patients were valid in the control arm, of whom 20 (69%) proceeded to surgery (RR 1.35; 95% CI 1.05–1.74). PCR was found in 37% of patients in the nivolumab and chemotherapy arm vs. 7% of patients in the control arm (RR 5.34; 95% CI 1.34–21.23). MPR were 53% vs. 14%, respectively, and ORR was 75% vs. 48% in each arm. Moreover, 24-month PFS was 67.2% (95% CI 55.8–81%) vs. 40.9% (95% CI 26.2–63.6%) in the experimental and control arms (HR 0.47; 95% CI 0.25–0.88), respectively. Additionally, 24-month OS was 85% (95% CI 75.9–95.2%) vs. 63.6 (95% CI 47.8–84.6%), respectively (HR 0.43; 95% CI 0.19–0.98). When stratifying these results based on pCR, PFS and OS at 24-month were 100% in the nivolumab and chemotherapy group, establishing pCR as a predictive value for PFS and OS. Grade 3 or greater adverse events were reported in 24% of patients in the experimental arm vs. 10% of patients in the control arm. Only 1 patient had an adverse event in the experimental arm that led to the cancelation of surgery.

The KEYNOTE-671 study [35] is a randomized phase III trial in which patients with resectable stage II, IIIA or IIIB (N2 stage) NSCLC were randomized 1:1 to receive neoadjuvant pembrolizumab or placebo, which was combined with cisplatin-based chemotherapy for four cycles occurring every 21 days, followed by surgery. Patients received adjuvant pembrolizumab or placebo once every 3 weeks for up to 13 cycles. The primary endpoints were EFS and OS. Secondary endpoints were MPR, pCR and safety. In total, 397 patients were randomized to the pembrolizumab group and 400 patients were randomized to the placebo group. At 24 months, the EFS was 62.4% in the pembrolizumab group and 40.6% in the placebo group (HR 0.58; 95% CI 0.46–0.72), while the OS was 80.9% and 77.6% of patients in each group, respectively. MPR occurred in 30.2% of patients in the pembrolizumab group and 11% of patients in the placebo group. pCR was observed in 18.1% and 4% of patients in each group, respectively. Grade ≥ 3 adverse events were reported in 44.9% of patients in the pembrolizumab group and 37.3% of patients in the placebo group.

## 4. Sex Differences in Immune Response

In recent years, sex has been identified as one of the main elements that can modulate the immune response [41,42]. Women usually elicit a stronger immune response than men [43], a fact that might explain why autoimmune diseases prevail in this group of patients, as well as why infections are more severe in the male population [39]. At the basic level, multiple differences have been found in the innate and adaptive immune systems of both sexes [44,45,46]. In adult humans, there are differences in sex lymphocyte subsets: there is a higher number of CD4+ T cell counts and higher CD4/CD8 ratios in females than in same-age males. Transcriptional analyses have also shown a higher cytotoxic T cell activity in females than in males [41]. All of these differences point to sex differences in the immune response triggered against NSCLC, which led to the complete analysis of this phenomena by Conforti et al. [47]. This analysis showed important differences in early-stage tumors (stage I–III). In a pooled analysis, it was found that dendritic cells, CD4+ T cells, B cells and Mast cells were more enriched in the tumor microenvironment (TME) of women than in men, with a false discovery rate cut-off ≤0.05. Other cells found to be relevant in this response were regulatory T cells, natural killer T cells, M1 type macrophages, CD8+ T cells and eosinophils. TME in female patients was also significantly enriched in cancer-associated fibroblasts, granulocyte–macrophage progenitors and hematopoietic stem cells, which have been shown to exert immunosuppressive activities in the TME. The T-cell landscape was also analyzed in early-stage tumors [47]. The following analyzed T-cell subpopulations were significantly enriched in the TME of women: (1) CD8+ and CD4+ naïve T cells, (2) CD8+ and CD4+ effector T cells, (3) CD8+ and CD4 T-cell subpopulations with an intermediate functional state. Previous studies showed that a higher clonality of T cell receptor (TCR) tumor-infiltrating lymphocytes (TILs) is an indirect comparator T-cell immune response to tumor antigens compared to a polyclonal TCR [48,49]. Conforti et al. found a significantly greater TCR clonality in the TILs of women [47]. Immune evasion mechanisms also differ based on sex. T-cell dysfunction has been shown to be higher in the tumors of women than in those of men [46]. In contrast, the mean value of the “T-cell exclusion” score always had higher results in the tumors of men. Significantly higher expression levels of the following inhibitory immune checkpoints in TME have been found in female patients: TIM3, TIGIT, BTLA, ADORA2A, ENTPD1, TNFRSF14, VISTA, BTN3A1. They are known to be key factors influencing T-cell exhaustion mechanisms and are explored as therapeutic targets [47,50,51].

Taking all of these results into account, like the lower number of immune cells in TME, the higher T-cell exclusion score, and the smaller TCR clonality, they show that tumors in men have less efficient tumor infiltration via the immune system and less tumor recognition [47]. However, because of the efficiency of the immune system of women, NSCLC develops more complex and redundant mechanisms of resistance, as shown by the higher expression of immune checkpoint molecules with inhibitory functions. A higher abundance of immune-suppressive cells in TME and Tregs is seen as well [50,51]

### Sex-Based Immune Response to ICI

Despite their being scarce literature addressing sex-based difference response to ICIs, some differences have been reported. Hormones can change the function and expression of PD-1 and mediate autoimmunity [52,53]. Furthermore, in pre-clinical melanoma murine models, the different efficacy of anti-PD-L1 in relation to the sex has been described [54]. In this sense, Conforti et al. [55] hypothesize that male patients could have a better benefit from ICI in metastatic disease than female patients due to three considerations: (1) There is sex dimorphism in immunity. Tumors in woman tend to have a better immune surveillance system and need a stronger immune-editing process to produce metastases [55]. This process could make the tumors less immunogenic and have more mechanisms to evade the immune system. This issue would make the tumor more resistant to immunotherapy [56]. (2) The tumor mutational burden is significantly higher in male patients, irrespective of other factors [57]. Finally, (3) there is a lower smoking prevalence in female than male populations, which affects the tumor mutational burden stemming from this behavior [58].

A metanalysis of immunotherapy in metastatic patients, including NSCLC and small-cell lung cancer (SCLC), included 20 randomized controlled trials (two phase II trials, 17 phase III trials, and one phase II-III trial) [55]. Seven trials were conducted in patients with melanoma, 6 trials were conducted in patients with NSCLC, 2 trials were conducted in patients with head and neck cancer, and 1 trial was conducted in patients with SCLC, renal cell carcinoma, gastric tumors, mesothelioma and urothelial tumors. Male patients treated with ICIs had a significantly reduced risk of death compared with men in control groups (pooled OS HR 0.72; 95% CI 0.65–0.79). In female patients, there was also a benefit when ICI were used, albeit to a lesser extent (pooled OS HR 0.86; 95% CI 0.79–0.93). Interestingly, there was a significant difference in the efficacy of ICIs between men and women compared to control groups of each sex (pooled interaction HR 0.85; 95% CI 0.77–0.94). Other metanalysis have tried to confirm this difference by including different immunotherapy studies [59,60]. However, one of the main conclusions of these studies is that the clinical trials used are incapable of exploring the effect of sex disparities due to heterogeneity in female patients (disparity in the number of patients), the lack of sex subgroup data, and the absence of mature OS and PFS. These data are completely aligned with our hypothesis stating that in females with metastatic NSCLC, an evasion of a more proficient immune system may limit the efficacy of ICI.

## 5. May Sex Limit Response to Neoadjuvant ICIs?

To answer this question, we used data from three single arm phase II trials and 1 randomized phase III trial that met our eligibility criteria, i.e., to generate results of MPR based on sex. The heterogenicity of MPR was not significant (*p* = 0.1; I^2^ = 52%), meaning that the fixed effects model was used. The MPR OR comparing males and females was 1.82 (95% CI 1.13–2.93; *p* = 0.01), which was statistically significant and favored the female subgroup (Figure 1). A second analysis was performed using the studies that reported pCR based on sex, which included three phase II trials (two single arm and one randomized) and one phase III trial. The heterogenicity of pCR was significant (*p* = 0.75; I^2^ = 0%), meaning that the random effects model was used. The pCR OR comparing males and females was 1.62 (95% CI 0.97–2.75; *p* = 0.08), which was not statistically significant, although there is a tendency to favor the female subgroup (Figure 2).

## 6. Discussion

The perioperative landscape of NSCLC is continuously evolving and will change the clinical practice of treating this disease in the coming years. The neoadjuvance was relegated to specific settings before the introduction of ICIs [21,22], but the evidence we reviewed shows a change in paradigm [24,25,26,27,28,29,30,31,32,33,34,35]. The neoadjuvant approach is quite promising, having an increased number of MPR and pCR [24,25,26,27,28,29,30,31,32,33,34,35], which has led to one of the most important changes in everyday clinical practice with the approval of the CheckMate-816 protocol by the U.S. Food and Drug Administration (FDA) and the European Medicines Agency (EMA). However, different approaches in many clinical trials are yet to show the optimal sequence of treatment, such as pre-operative ICIs versus adjuvant ICIs. Moreover, there are still some questions that need to be answered: Might the surgical procedure become more difficult for the thoracic surgeons due to treatment-induced changes? What is the optimal stage at which to use this treatment? Should all patients receive a combination of ICIs plus chemotherapy, or are there some settings in which ICI monotherapy could be enough? According to evidence of the phase III CheckMate-816 and KEYNOTE-671 trials, it seems that surgical complications are the same or less frequent in the ICI plus chemotherapy group than in the chemotherapy alone group [32,35]. Regarding the optimal stage to select, the phase III CheckMate-816 and KEYNOTE-671 trials [32,35], which included stage IB (in the CheckMate-816 trial), II, and III patients, showed a global benefit for all of the cohort. However, the benefit was higher in patients with stage IIIA tumors. This effect led studied authors to establish in the SAKK 16/14 [30], NADIM [31] and NADIM-II [33,34] trials an inclusion criterion, which required the involvement of stage IIIA patients. This point is controversial, as it is argued in the discussion of the CheckMate-816 and KEYNOTE-671 trials that the stage IB or II patients were under-represented, and further research should be performed in that specific setting. Future approaches could include a comparison of neoadjuvant combination of ICIs plus chemotherapy with adjuvant ICIs, like atezolizumab [61], in different stages (e.g., IB, II, III). Nevertheless, it seems clear that, with ICIs, it is possible to rescue many stage IIIA patients who would not be operated on in the past, which is a great achievement at such a poor prognostic stage. Another setting that would be interesting to compare is the use of neoadjuvant ICIs, followed by surgery against chemotherapy plus radiotherapy, as a radical treatment in the stage IIIA setting [62], particularly in case of N2 disease. Future treatment approaches that could modulate the tumor immune environment are currently being studied, such as the use of metformin-modified chitosan to increase the susceptibility of platin-based chemotherapy and downregulate PD-L1 expression [63]. Regarding the optimal regimen and settings that need to be selected, including the addition of chemotherapy, different clinical factors should be considered: One of the most relevant factors, which is obviated in many trials, is sex. Historical series show a better outcome of NSCLC in woman than in men [64,65], and part of this difference could be explained based on the better immune control of the disease in women in the localized setting.

As we have found, sex is a key actor that exerts influence over the immune response to NSCLC and other tumors. Here, we show that sex can influence responses to ICIs in the neoadjuvant setting (Figure 3). In our two small meta-analysis, we found that female patients responded better to ICIs in monotherapy or when combined with chemotherapy. In the first method (Figure 1), which included MPR data stratified based on sex, the result was statistically significant, with a clear tendency toward a higher number of MPR in this cohort. When analyzing the trials that included pCR stratified based on sex (Figure 2), although the result was not statistically significant, there is a clear tendency that favors women. The cause of not reaching the statistical significance level may be the lack of potency and the heterogeneity of the trials (a random effects model was required), which might be solved using further data from future trials. Other biases affecting our study are the lack of sex data provided in some of the reviewed trials, the absence of sex as a main stratification factor in all of the reviewed trials, and the scarce number of trials available to analyze. Nevertheless, on a global level, these results support our hypothesis. Women have a more proficient immune system [43], which might grant them better control of localized NSCLC than men and a better response to ICIs. However, this situation changes with metastatic disease, as Conforti et al. [55] showed: localized NSCLC needs to escape a more proficient immune system in women to become metastatic, meaning that it might be more resistant to ICIs; on the other hand, men would respond better to ICIs when NSCLC becomes metastatic, since it escaped a less proficient immune system. This effect, as has been seen in the metastatic NSCLC, can be alleviated through the use of a combination of chemotherapy and ICIs, since their synergistic effects, such as the generation of neoantigens, may help to induce an immune response [66]. This effect complies with the results of the Checkmate-816 and KEYNOTE-671 trials, which seem to show fewer differences between men and women [32,35]. Taking this data into account, it would be interesting to investigate in the future if ICIs without chemotherapy could be sufficient for the NSCLC neoadjuvant setting in female patients.

ICIs are improving the state of the art of the perioperative treatment in NSCLC, as the promising results of the reviewed phase II and III trials show. However, there are many doubts regarding the optimal scenario and use of these treatments: one of the main factors that should be considered is sex, and future investigations should explore its influence to establish the optimal regimens for each situation.

## 7. Conclusions

ICIs have changed the paradigm of the neoadjuvant treatment of NSCLC by increasing MPR and pCR, which was unattainable through the use of chemotherapy alone, a fact that led to the neoadjuvant setting being discarded barring some specific situations. Yet, there are many doubts regarding the optimal combinations of drugs that should be used, the stages that benefit from this approach and the relevant factors that should be considered. Data are scarce, but as our analysis shows, sex is a key element. It is known that women have a more proficient immune system, which may help to control disease in localized NSCLC. This fact could enable disease control in non-metastatic NSCLC and induce a better response to ICIs compared to men, most likely due to women’s superior immune capacity to detect and remove tumor cells. Here, we show that sex has influence over the immune checkpoint inhibitor’s response in the neoadjuvant setting by showing more benefit in women than in men. Our results, which need to be confirmed when further data appear, suggest that in the future, ICIs neoadjuvant regimens should be personalized based on sex and ICI monotherapy could be more appropriate for women, while combination of chemotherapy and ICIs could be more convenient for male patients.

## Figures and Tables

**Figure 1 cancers-15-04433-f001:**
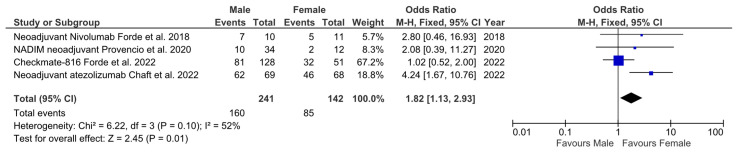
Forest plot of the MPR stratified based on male and female of the trials with available sex data [24,26,31,32].

**Figure 2 cancers-15-04433-f002:**
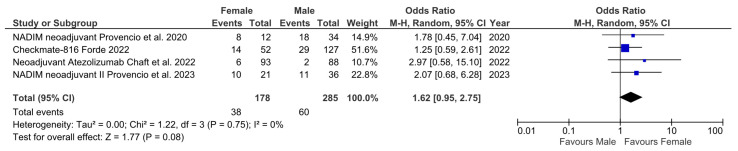
Forest plots of the pCR data stratified based on male and female sex derived from the trials using available sex data [26,29,32,34].

**Figure 3 cancers-15-04433-f003:**
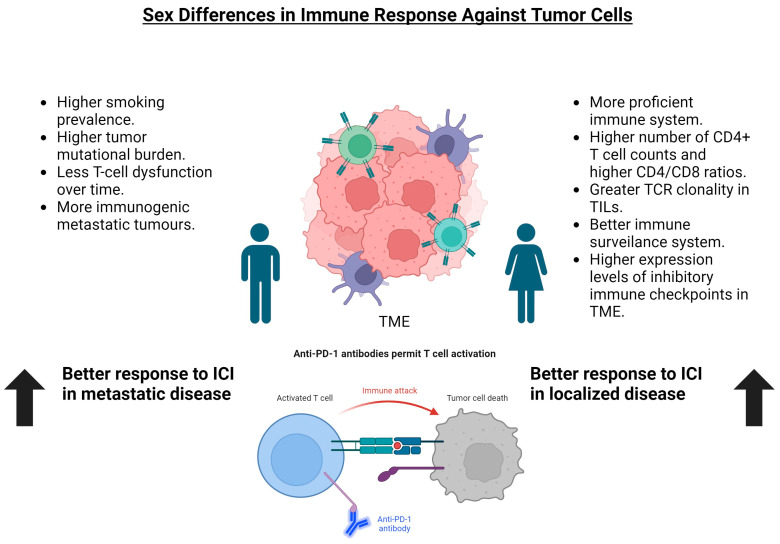
Summary of the main differences between the immune responses of males and females to NSCLC tumor cells. A more proficient immune system in women sets a better immune response to the tumor cells in the initial stages of disease. However, tumor cells acquire more resistance mechanisms to act against the immune response over time, meaning that the metastatic setting immune cells in women do not respond as efficiently as they do in men. In the case of men, the immune system is not as proficient, and tumor mutational burden is higher due to factors like higher tobacco consumption. Those factors mean that the tumor requires less immune evasion mechanisms to become metastatic, but suggest that the TME becomes more immunogenic and develops a more favorable response to ICIs. TCR: T-Cell Receptor; TIL: tumor infiltration lymphocyte; TME: tumor microenvironment; ICI: immune checkpoint inhibitor.

**Table 1 cancers-15-04433-t001:** Summary of the clinical trials that used ICI and recorded MPR and pCR. MPR: major pathological response; pCR: pathological complete response.

Study	Regimen	Phase	Patients	MPR	pCR
Forde et al., 2018 [24] NCT02259621	Nivolumab	II	20	45%	15%
Wislez et al., 2020 [28] NCT03030131	Durvalumab	II	46	(Study finished early)	(Study finished early)
Besse et al., 2020 [27] NCT02994576	Atezolizumab	II	29	0%	0%
Cascone et al., 2021 [25] NCT03158129	Nivolumab vs. Nivolumab and Ipilimumab	II	44	24% vs. 50%	10 vs. 38%
Chaft et al., 2022 [26] NCT02927301	Atezolizumab	II	143	20%	6%

**Table 2 cancers-15-04433-t002:** Summary of the clinical trials which included Chemotherapy plus ICI with results of MPR and pCR. MPR: major pathological response; pCR: pathological complete response.

Study	Regimen	Phase	Patients	MPR	pCR
Shu et al., 2020 [29] NCT02716038	Chemotherapy and Atezolizumab	II	30	57%	3%
Provencio et al., 2020 [31] NCT03081689	Chemotherapy and Nivolumab	II	46	83%	63%
Rothschild et al., 2021 [30] NCT02572843	Chemotherapy and Durvalumab	II	67	62%	18%
Provencio et al., 2022 [33,34] NCT03838159	Chemotherapy and Nivolumab vs. chemotherapy	II	57 vs. 29	53% vs. 14%	37% vs. 7%
Forde et al., 2022 [32] NCT02998528	Chemotherapy and Nivolumab vs. chemotherapy	III	179 vs. 179	36.9% vs. 8.9%	24% vs. 2.2%
Wakelee et al., 2023 [35] NCT03425643	Chemotherapy and Pembrolizumab vs. Chemotherapy	III	397 vs. 400	30.2% vs. 11%	18.1% vs. 4%

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
