# Peer review of "Sex Differences in the Efficacy of Immune Checkpoint Inhibitors in Neoadjuvant Therapy of Non-Small Cell Lung Cancer: A Meta-Analysis"

_cancers, 2023, doi:10.3390/cancers15184433_

Round 1

Reviewer 1 Report

The study is important and interesting. The authors should clarify in the title and abstract that this paper is a meta-analysis.

How did the authors deal with the high heterogeneity observed in their meta-analysis? Did the authors try any forms of sensitivity analysis?

THe authors should comment more on the safety profile of these drugs, with particular reference to the risk of reactivation of immune disease (cite the recent MA: PMID: 33314269 )

Did the authors check the risk of publication bias?

Did the authors assessed the quality of the studies?

Author Response

Response to reviewer 1 comments:

The study is important and interesting. The authors should clarify in the title and abstract that this paper is a meta-analysis.

We thank the reviewer comments. We agree that this manuscript is not a “canonical” review since we have also carried out data analysis. In fact, the suggestion to classify it as meta-analysis has been also pointed by another reviewer and we agree with them. We have many doubts concerning how to classify the work when we submitted it. Anyway, we leave to editor’s consideration whether to classify it as review or meta-analyses manuscript. Following the reviewer’s suggestion, we have changed the type of paper and partially rewritten both the summary and the introduction of the article putting more emphasis in the focus of the paper and the meta-analysis (lines 27-29; 106-112).

How did the authors deal with the high heterogeneity observed in their meta-analysis? Did the authors try any forms of sensitivity analysis?

As already mentioned in the paper, there are only a limited number of studies about neoadjuvant setting in NSCLC, and unfortunately, not all of them report information regarding sex. In fact, this was one of the reasons that make this review timely and motivated us to carry out it. We agree that heterogeneity delineates a factor that has to be considered, thus, to address it, we performed the X2 and I2 tests in both meta-analysis (more information is available in the statistical analysis part in lines 140-147). In case heterogeneity was significant (e.g., pCR analysis), we decided to use the random effects model to attenuate it, despite, it implied that statistical significance was not obtained since the random effects model widens the confidence interval. In the MPR analysis, since heterogeneity was not significant, we used the fixed effects model.

The authors should comment more on the safety profile of these drugs, with particular reference to the risk of reactivation of immune disease (cite the recent MA: PMID: 33314269 )

To further clarify this aspect, we have now added a new paragraph in the introduction addressing the toxicity profile of activation/reactivation of autoimmune diseases (lines 79-85). We have also included the suggested cite (line 82). Furthermore, when each clinical trial is commented information regarding toxicity profile is briefly provided and properly referenced. We consider this information enough considering the scope of our work. Moreover, the two other reviewers have asked us to short the introduction.

Did the authors check the risk of publication bias?

Yes, we have considered, and we are aware of it and mentioned in the paper. Publication bias was assessed by means of a funnel plot in both analyses. In our analysis, we have the problem that not all the authors included sex data, thus, this data was not considered, which may introduce bias in the study. This limitation is commented as one of the weaknesses of our study (lines 444-446). In order to prevent publication bias as much as possible we also considered other communication sources, such as posters or small oral communications in congresses, so that even trials who were deemed negative and were not published as articles (such as the IONESCO or PRINCEPS trials) were included, which in our opinion strengths the manuscript. With the available data, our intention was just to generate curiosity in the scientific community that motivated researchers to conduct new trials considering sex, to then, repeat data analysis which most likely will further confirm our results.

Did the authors assessed the quality of the studies?

Yes, we attempted to. In our literature review process, the quality of the studies was assessed by means of the Preferred Reporting Items for Systematic Reviews and Meta-analysis (PRISMA). This procedure ensures that all the selected studies fulfil our inclusion criteria (e.g., sex data…). Because the number of trials was small, all the authors were able to analyse all the available trials so that they complied with the established cohort. We have added the used PRISMA methodology in the “Methods” section of the paper (lines 126-127).

Reviewer 2 Report

The Authors present a very noticeable review and meta-analysis of sex difference in efficacy of ICI administered as neoadjuvant therapy in NSCLC. The findings, although preliminary, may have an impact for clinical practice and pave the way for future research.

My main - and only - comment is on the structure of the paper. It is somehow unusual, one may expect a more "traditional" IMRAD structure, but it flows. Can however the Authors revise the Introduction to shorten the information on NSCLC and better clarify the aims and scopes of this analysis?

Author Response

Response to reviewer 2 comments:

The Authors present a very noticeable review and meta-analysis of sex difference in efficacy of ICI administered as neoadjuvant therapy in NSCLC. The findings, although preliminary, may have an impact for clinical practice and pave the way for future research. My main - and only - comment is on the structure of the paper. It is somehow unusual, one may expect a more "traditional" IMRAD structure, but it flows. Can however the Authors revise the Introduction to shorten the information on NSCLC and better clarify the aims and scopes of this analysis?

We acknowledge the positive comments of the reviewer. Initially this work was conceived as a formal review, but we always questioned ourselves whether sex could influence response to ICIs in the neoadjuvant setting as observed in other clinical settings, which is most likely due to the differences of male and female immune systems. Thus, to answer that question, we decided to perform a meta-analysis with the limited data available. Despite the obtained results strengthen the work, they brought us the doubt whether to classify the manuscript as a review or as meta-analysis. The limited data available to properly perform a reliable meta-analysis led us to classify the work as a review. However, after reading all the comments from the reviewers and if the editors agree with, we have decided to change the type of paper to meta-analysis. In order to reinforce it, we have abbreviated the first part about NSCLC to reduce the introduction (although we have expanded some elements which were suggested by other reviewers, such as the toxicity profile), and we have specified more clearly the main aims and scopes of the analysis (lines 106-112).

Reviewer 3 Report

       In this research, the authors reviewed the “Sex Differences in the Efficacy of Immune Checkpoint Inhibitors Neoadjuvant Therapy of Non-Small Cell Lung Cancer”. In my opinion, the current stage of this paper could meet the requirements of Cancers after major revisions.

My comments are as details:

1.      In Line 56-88, the current development and usage of PD-L1 inhibitors should be more carefully discussed. Some references should be added to this part including 10.1016/j.ijbiomac.2022.10.167.

2.      Some minor mistakes exist in the format of the references. The authors should carefully check it.

3.      The conclusion part was too plain. An in-depth outlook or conclusion should be added.

4.      Some references are out of date, such as ref 55 and ref 60. Recent published papers may be better.

5.      The quality of Figure 1 and Figure 2 should be improved.

6.      In my opinion, more figures or tables that summarize the pre-clinical researched status of immune checkpoint inhibitors neoadjuvant therapy of non-small cell lung cancer should be added.

7.      The recent research that summarize the developments or effects the sex differences of immune checkpoint inhibitors neoadjuvant therapy of non-small cell lung cancer on T cell infiltration and its activity, as well as macrophage polarization and neutrophile granulocyte activity. Some related research was listed as below: 10.1016/j.jconrel.2022.11.004.

Author Response

Response to reviewer 3 comments:

  1. In Line 56-88, the current development and usage of PD-L1 inhibitors should be more carefully discussed. Some references should be added to this part including 10.1016/j.ijbiomac.2022.10.167.

To clarify this aspect that is in line with the comments of other reviewers, we have changed some parts of the introduction and the discussion (lines 51-60, 79-85, 425-427). However, the limited space available restricts to further extend the use of PD-L1 inhibitors, which somehow is out of the scope of our work, and it has been more deeply addressed in other reviews. For these reasons and following the reviewer’s suggestion, we have added the bibliographic reference number 4 to refer readers to further information about the mechanism of action and pre-clinical knowledge of the immune checkpoint inhibitor drugs. (Doi: 10.3389/fphar.2021.731798). Additionally, we have briefly introduced the mechanism of action of the suggested treatments, we have also mentioned every phase III trial that includes immune checkpoint inhibitors (even in the metastatic setting, so this part should not be extended). We also considered that the mentioned trials are recognized by the guidelines of the main international clinical practice oncology associations (ESMO, ASCO, NCCN…). We agree that the basic part of drugs that modulate the tumour immune environment is too short, but since the scope is to analyse information from a clinical point of view, only approved immune checkpoint inhibitors (for now, only anti-PD1 and anti-CTLA4 drugs) have been considered. Future articles could focus on the pre-clinical bases of sex differences at a molecular level, which would be very relevant. Lastly, some part of that pre-clinical investigation, such as the reference: 10.1016/j.ijbiomac.2022.10.167 seems very interesting and promising, so we have added it as a “Future investigational approach” in the discussion (lines 425-427).

  1. Some minor mistakes exist in the format of the references. The authors should carefully check it.

References have now been checked and corrected to comply with the MDPI journals guide:  https://mdpi-res.com/data/mdpi_references_guide_v5.pdf

  1. The conclusion part was too plain. An in-depth outlook or conclusion should be added.

The conclusion has been expanded to better explain the concept, implications, practice-changing recommendations, and future investigations guidance (lines 476-490).

  1. Some references are out of date, such as ref 55 and ref 60. Recent published papers may be better.

We agree and therefore reference 55 has been updated to the 2021 MPOWER WHO report. Reference 60 and 61 were selected as “historical” references to exemplify and reinforce the idea that sex has been proven to modulate response in the treatment of NSCLC not only with ICIs but also with other therapies. Therefore, we have decided to maintain them.

  1. The quality of Figure 1 and Figure 2 should be improved.

Figure 1 and Figure 2 quality has been improved. The file has been imported, as a Scalable Vector Graphics file (SVG), which should prevent quality loss one copied into the article.

  1. In my opinion, more figures or tables that summarize the pre-clinical researched status of immune checkpoint inhibitors neoadjuvant therapy of non-small cell lung cancershould be added.

Regarding the pre-clinical data and in line with our previous comments, we think that a more profound discussion of pre-clinical aspects would expand the paper too much and would deviate its focus, which is mainly centered in the clinical practice. However, following the reviewer suggestion we have now designed a very simple scheme that summarizes basic aspects (see Figure 3), and we have also introduced new references to refer interested readers to such kind of information.

  1. The recent research that summarize the developments or effects the sex differencesof immune checkpoint inhibitors neoadjuvant therapy of non-small cell lung cancer on T cell infiltration and its activity, as well as macrophage polarization and neutrophile granulocyte activity. Some related research was listed as below: 10.1016/j.jconrel.2022.11.004.

We have now included a small summary (Figure 3) of the physiological bases of sex differences in the response of immune checkpoint inhibitors, which hopefully will briefly clarify basic aspects and improve the manuscript´s focus despite it was not the scope of the paper, the space limit, and the comments from the two other reviewers who recommended us to reduce the introduction. Additionally, we have increased the number of bibliography citations in order to allow the readers to further expand their pre-clinical knowledge, for instance, cite 42 (doi: 10.3390/biomedicines8070232).

Round 2

Reviewer 1 Report

The revised version of the manuscript is OK. Thank you!

Reviewer 3 Report

The current version of this manuscript could be accepted.